# A data-driven characterisation of natural facial expressions when giving good and bad news

**David M. Watson** *, **Ben B. Brown, Alan Johnston**

School of Psychology, University of Nottingham, Nottingham, United Kingdom

* david.watson@nottingham.ac.uk

**Data Availability Statement:** The code and materials are publicly available on the Open Science Framework (https://osf.io/6tbwj/).

**Funding:** This research was funded by the NIHR Nottingham Biomedical Research Centre and

## Abstract

Facial expressions carry key information about an individual's emotional state. Research into the perception of facial emotions typically employs static images of a small number of artificially posed expressions taken under tightly controlled experimental conditions. However, such approaches risk missing potentially important facial signals and within-person variability in expressions. The extent to which patterns of emotional variance in such images resemble more natural *ambient* facial expressions remains unclear. Here we advance a novel protocol for eliciting natural expressions from dynamic faces, using a dimension of emotional valence as a test case. Subjects were video recorded while delivering either positive or negative news to camera, but were not instructed to deliberately or artificially pose any specific expressions or actions. A PCA-based active appearance model was used to capture the key dimensions of facial variance across frames. Linear discriminant analysis distinguished facial change determined by the emotional valence of the message, and this also generalised across subjects. By sampling along the discriminant dimension, and back-projecting into the image space, we extracted a behaviourally interpretable dimension of emotional valence. This dimension highlighted changes commonly represented in traditional face stimuli such as variation in the internal features of the face, but also key postural changes that would typically be controlled away such as a dipping versus raising of the head posture from negative to positive valences. These results highlight the importance of natural patterns of facial behaviour in emotional expressions, and demonstrate the efficacy of using data-driven approaches to study the representation of these cues by the perceptual system. The protocol and model described here could be readily extended to other emotional and non-emotional dimensions of facial variance.

## Author summary

Faces convey critical perceptual information about a person including cues to their identity, social traits, and their emotional state. To date, most research of facial emotions has used images of a small number of standardised facial expressions taken under tightly controlled conditions. However, such approaches risk missing potentially important facial signals and within-person variability in expressions. Here, we propose a novel protocol that allows the eliciting of emotional expressions under natural conditions, without

supported by the NIHR Nottingham Clinical Research Facilities. The views expressed are those of the authors and not necessarily those of the NHS, the NIHR, or the Department of Health and Social Care. The funder had no role in study design, data collection and analysis, decision to publish, or preparation of the manuscript.

**Competing interests:** The authors have declared that no competing interests exist.

requiring people to deliberately or artificially pose any specific facial expressions, by video recording people while they deliver statements of good or bad news. We use a model that captures the key dimensions of facial variability, and apply a machine learning algorithm to distinguish between the emotional expressions generated while giving good and bad news. By identifying samples along the discriminating dimension and projecting them back through the model into the image space, we can derive a behaviourally relevant dimension along which the faces appear to vary in emotional state. These results highlight the promise of data-driven techniques and the importance of employing natural images in the study of emotional facial expressions.

## Introduction

Human faces convey a wealth of person-specific information. For instance, they offer salient cues to a person's identity [1], allow the inference of key social traits [2], and provide critical information about an individual's emotional state [3,4]. The perceptual system is able to efficiently extract such information despite substantial variability in facial features both between and within individuals, for instance resulting from changes in viewpoint or environmental conditions [5].

Models of face perception generally propose two key accounts of how faces convey emotional information. Categorical models posit that facial expressions can be assigned to a series of discrete emotion classes [6]. Particular focus has been given to six or seven basic emotion categories that are considered culturally universal [7,8], though the universality of some of these categories has been disputed [9]. By contrast, continuous models propose that facial expressions can be represented as points varying along multiple emotional continua—such as valence or arousal—within a multi-dimensional face space [10,11]. For instance, expressions of happiness and sadness could be represented as two discrete and independent classes under a categorical model, or as projections along opposing directions of a single valence dimension under a continuous model. Categorical accounts are supported by the fact that facial expressions from basic emotion categories are easily recognised [6], and that expression changes are often perceived categorically [12]. However, categorical models struggle to account for subtler or more nuanced expressions that diverge from classic universal categories, or why some categories (such as fear and surprise) are frequently confused for each other. Such cases may be more readily explained by a continuous account.

To date, research into facial expressions has almost ubiquitously employed stimuli generated under tightly controlled experimental conditions. Images in face databases are often rigidly controlled to standardise facial expression and head pose, as well as environmental factors such as lighting. The aim of such approaches is to minimise extraneous sources of image variation as far as possible, allowing for greater focus on key variations in facial features. However, such approaches also pose a number of critical limitations. Firstly, applying such extreme controls necessarily adds a degree of artificiality to such images. In attempting to minimise sources of extraneous variation such approaches risk also eliminating informative dimensions of facial variation, such as within-person variability [5]. Indeed, so-called *ambient* face images, which are left deliberately uncontrolled to retain more natural variability, have been shown to yield critical information for face identification [5,13] and perception of social inferences [14] that is not present in traditional controlled stimuli. Secondly, the selection of which emotional categories and expressions should be posed is necessarily a subjective one. For instance, particular focus is often given to a small number of basic universal emotions [6]. However, this risks

biasing the dimensions of facial variation according to preconceived notions on the part of the experimenter as to which sources of emotional variation should be included. Finally, the majority of face databases employ static images, which fail to capture potentially informative sources of dynamic variation present in moving faces [15]. Thus, the extent to which the patterns of variance in facial expressions observed in traditional artificially-controlled static-images might generalise to more natural dynamic faces remains unclear.

More nuanced expressive facial stimuli may be generated by instead sampling emotions that vary along an emotional continuum. One common approach to this is to visually morph between two images depicting different categorical expressions [12,16]. A more data-driven method was applied by Calder and colleagues [17], who derived a principal components representation of several face images spanning multiple basic emotion categories, then applied linear discriminant analysis to discriminate the emotion categories. Not only could categories be accurately discriminated, but back-projecting the discriminant functions into the image space revealed critical dimensions of expression variation. However, all such techniques still typically use traditional artificially-controlled images as inputs to the procedure. Thus, the relationship between nuanced emotional continua and natural patterns of facial variation still remains unclear.

Here, we address previous limitations by advancing a novel protocol for eliciting emotional expressions from dynamic and natural facial behaviour, using a dimension of emotional valence as a test case. We employ a data-driven approach to extract emotional facial cues from these natural behaviours, without requiring subjects to deliberately or artificially pose any specific expressions. Subjects delivered short sentences to camera conveying either positive or negative news. The principal dimensions of facial variation were captured by representing each facial expression image by the vector field required to register the face to a reference expression plus the facial texture warped by this vector field. Registration was accomplished by a two frame version of the multi-channel gradient motion model [18–20]. A machine learning algorithm (Linear Discriminant Analysis; LDA) was then used to derive a continuous emotional dimension from the subjects' natural facial variability. This technique was not only able to successfully discriminate the emotional state of the faces, but also derive a behaviourally interpretable dimension of emotional valence.

## Results

### Recordings

Three subjects delivered a series of phrases to a video camera, conveying either positive (e.g. "Good news–you've got the job") or negative news (e.g. "I'm sorry to say the operation didn't go well"). Subjects each delivered 10 unique positive and 10 unique negative phrases, with Subjects 1 and 2 performing 15 repeats (300 phrases total) and Subject 3 performing 16 repeats (320 phrases total) of each phrase. A full list of the phrases is provided in S1 Table. Subjects viewed and addressed a series of putative recipients (consisting of listeners in a YouTube recorded video conversation) while performing the repeats to increase variability in expressing the message. Subjects were instructed to deliver the phrases in whatever they felt was the most natural style to them. Importantly, they were not asked to deliberately pose any specific facial expressions or perform any other particular actions.

For further analysis, we cut each clip to one of two time periods. *Prefix* clips were generated by clipping to the initial 1.44s (36 frames) after speech onset, thereby restricting each clip primarily to the "Good news" or "I'm sorry to say" prefix portion of the phrase. However, these clips present a potential confound in the form of the common linguistic information shared across clips. To address this, we also generated *suffix* clips in which phrases were cut to exclude

the initial common prefix portions and instead include only the latter unique suffix portions of each phrase (e.g. "you've got the job" or "the operation didn't go well").

## Multi-channel gradient model

We used an active appearance model, exploiting the Multi-channel Gradient Model (McGM) [18–20] for registration, to capture the key dimensions of facial variability over frames (Fig 1A). This technique has previously been shown to be successful in extracting and describing multiple dimensions of facial variance [20–22]. This model warps each frame to best align the facial texture to a single reference image extracted from the video sequence. Each frame is then represented as a 5-channel image comprising the RGB image texture information warped to the reference, and the x- and y-direction warp components. Each frame is thus described in terms of a "shape-free" version of its textures and the motion vectors that need to be applied to warp the original textures to the reference. Flattening each frame to a vector allows representing it in a high-dimensional feature space, with each dimension representing a specific pixel and channel, and with samples comprising each of the individual frames concatenated across clips (Fig 1B). The dimensionality of this space was reduced via Principal Component Analysis (PCA), retaining sufficient components to explain 90% of the variance across samples. PCA-based face spaces have previously been demonstrated to capture numerous key dimensions of facial variation [4,23] including emotional expressions [17].

Next, we used Linear Discriminant Analysis (LDA) to classify the positive versus negative valence phrases based on their representations within the McGM-PCA space [17]. We first

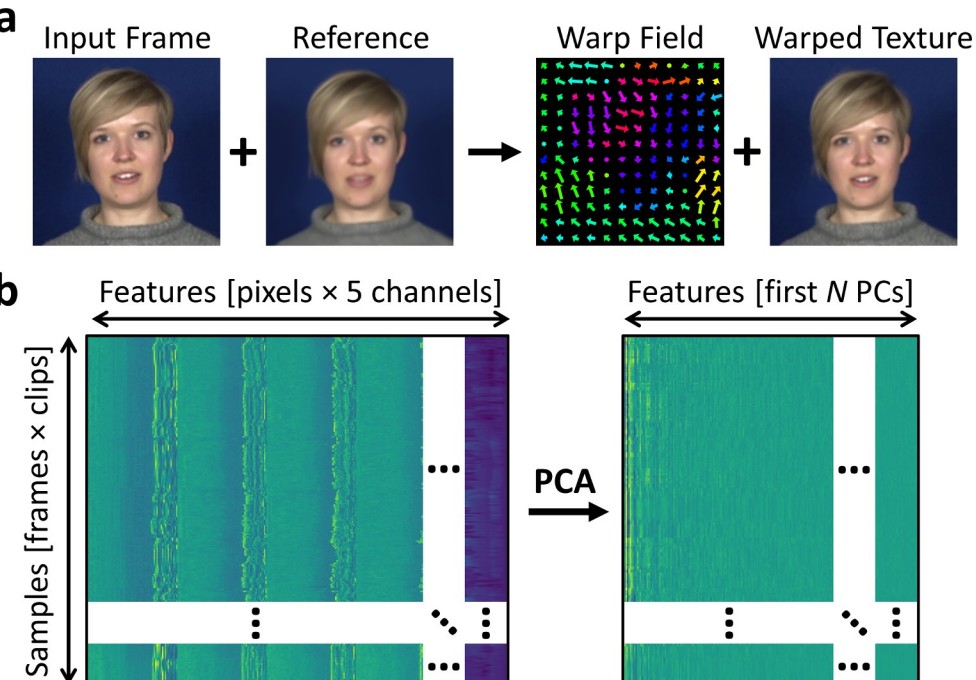

**Fig 1. McGM-PCA pipeline.** (a) Illustration of the McGM process for an example frame. Each frame is warped to a common reference, such that it is represented by a 5-channel image containing the x- and y- warp components and RGB warped textures. (b) Each warped frame is flattened to a vector, such that frames are represented in a high-dimensional feature space defined by the pixels and 5 image channels. The dimensionality of this space is reduced via principal component analysis, retaining sufficient components to explain 90% of the variance.

examined the cross-validated classification accuracy to ensure the valences could be discriminated reliably. We employed a (10-fold) leave-one-phrase-out cross-validation scheme. Within each subject independently, the LDA algorithm was fit to all samples for 9 out of the 10 phrases in each class, and then tested on all samples for the held-out phrases. This procedure was then repeated such that every phrase in each class was assigned as the held-out phrase once. The leftmost column of Fig 2 shows the resulting confusion matrices averaged over cross-validation folds and subjects: correct classifications are represented on the diagonal, while off-diagonal elements indicate misclassifications. A per-subject summary of the classification accuracies (given by the average of on-diagonal matrix elements) is shown in Fig 3. Near perfect classification was achieved: accuracies appeared close to ceiling and were significantly greater than chance in all subjects (one-sample t-tests: all $p < .001$). Importantly, this result was observed for both the *prefix* and *suffix* clips, indicating that classification performance generalises beyond the linguistic commonalities present in the initial prefix portions of each phrase.

To probe the stimulus features underlying the discrimination performance, we identified samples along the discriminant dimension and back-projected them into the image space. As this analysis does not require cross-validation we re-fit the LDA algorithm to the full dataset (including all 10 phrases). We identified the discriminant dimension as the line lying orthogonal to the decision boundary and passing through the centroids of each class. We then projected samples between ±3 standard deviations along this dimension. By inverting the PCA these samples were transformed from the McGM-PCA space to the McGM space, then an inverse warp further transformed them to the original image space. Examples of samples at 0, ±1.5, and ±3 standard deviations are shown in Fig 4. Video animations of the projections are also supplied in the supporting information (S1–S6 Videos). Note that these images are not frames that were ever actually acquired from the subjects–rather they represent hypothetical images simulated by projecting samples through the feature spaces. The negative versus

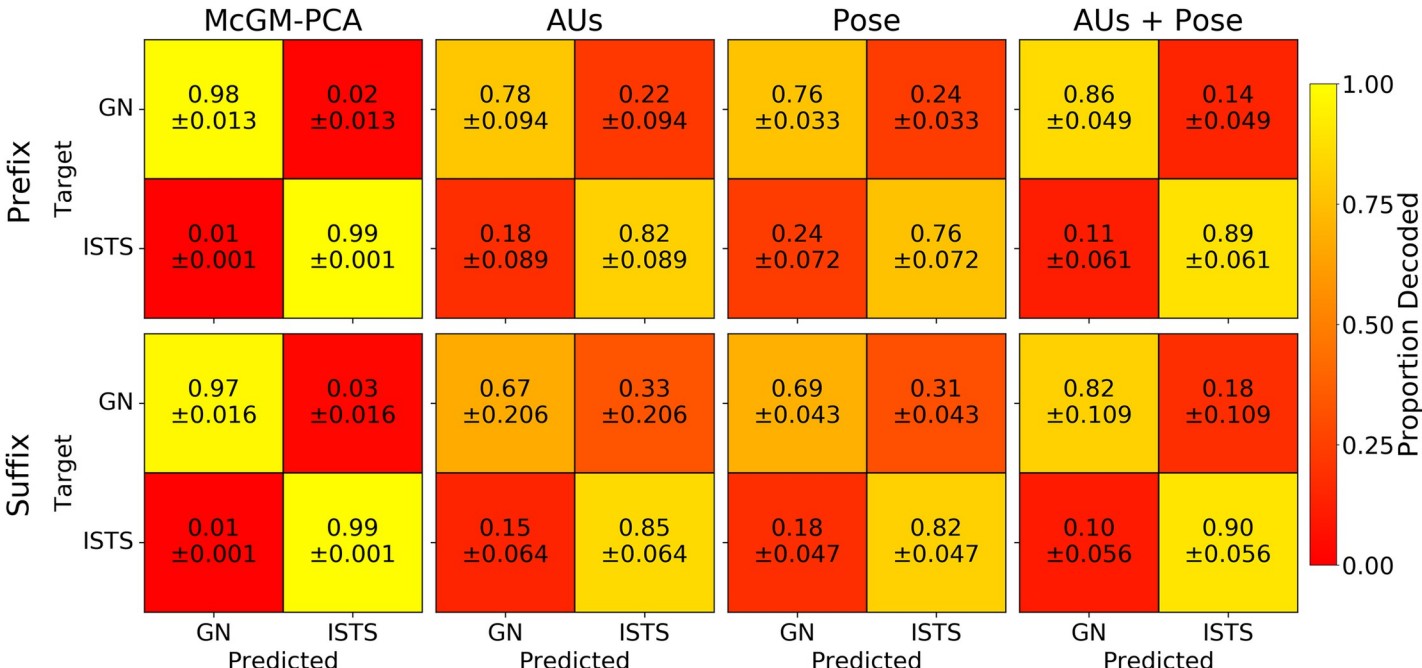

**Fig 2. Within-subject cross-validated confusion matrices.** Matrices indicate mean proportion of LDA classifications for each combination of "good news" (GN) and "I'm sorry to say" (ISTS) classes. Models are displayed across columns, and top and bottom rows display results for prefix and suffix clips respectively. Annotations indicate means and standard deviations of values over subjects.

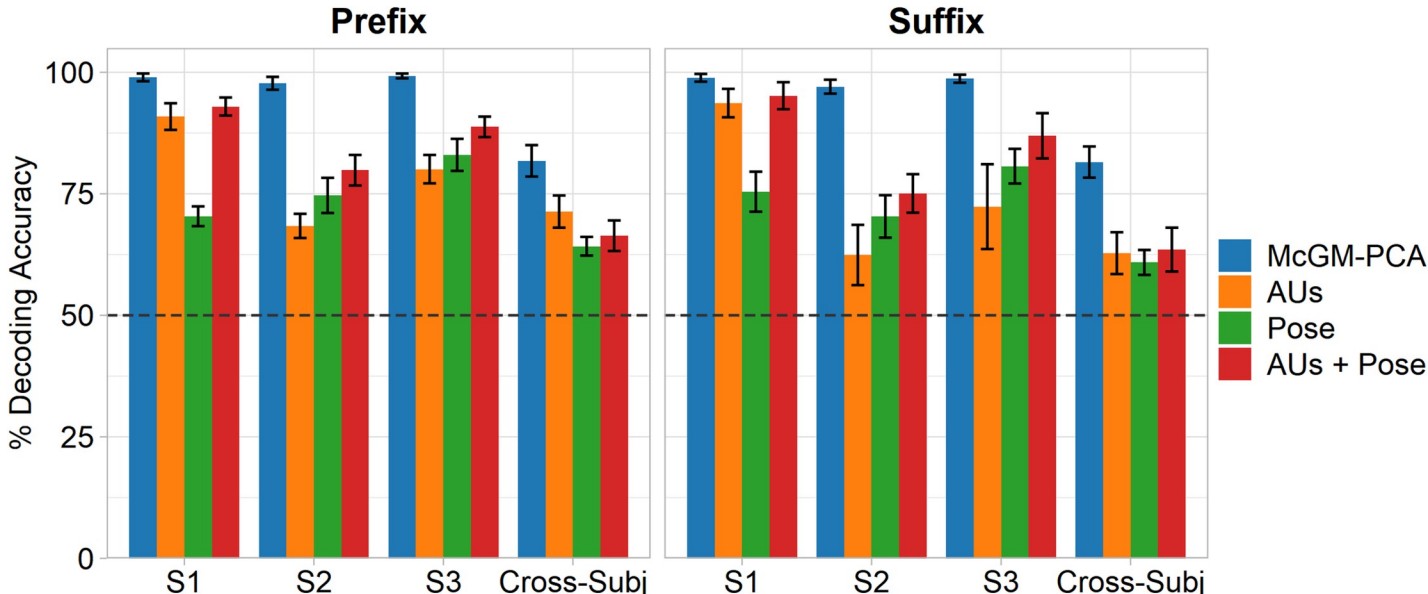

**Fig 3. Summary of cross-validated classification accuracies.** LDA classification accuracies for each model across subjects and for the cross-subject analyses. Values are averaged over cross-validation folds. Error bars indicate the standard error of the mean. The dashed line indicates the 50% chance level.

positive valence of these faces at either end of the dimension is clearly evident for all subjects and for both *prefix* and *suffix* clips. Intermediate samples illustrate gradations along a continuous dimension of emotional valence. While some of the facial variations along this dimension are captured by changes in facial expression typically represented in face databases (such as changes in the internal features), there are also other changes in external features (such as head pose) that would often be controlled away in standardised images. The patterns of variation observed in each subject appear highly similar between *prefix* and *suffix* clips, suggesting the discriminant dimension is largely independent of linguistic cues.

To quantify this relationship more formally, we extracted features from the back-projected images following the Facial Action Coding Scheme (FACS) [24,25]. Thirteen images were sampled along the discriminant dimension (±3 standard deviations in steps of 0.5). We then used the OpenFace toolbox [26,27] to automatically extract head pose (position and orientation) and intensities of 17 action units for each of these images. The relationship between each of these features and the LDA projections are shown in Fig 5 and Fig 6 for *prefix* and *suffix* clips respectively. The most prominent source of variation is in the vertical position and pitch of the head pose, corresponding to a dipping / raising of the head from negative to positive valence. Subjects 1 and 3 also show a clear modulation of head yaw (left/right turn of the head) but in opposing directions, while all subjects show modulations of head roll (cocking head to one side) but to different magnitudes and again often in different directions. Action units 1 (inner brow raiser), 12 (lip corner puller), and 17 (chin raiser) are broadly consistent over subjects in terms of the direction of the effect, though are more variable in terms of magnitude and rate of change. Other action units are less consistent; for instance, AU2 (outer brow raiser) is negatively associated with valence in Subject 1, but positively associated in Subject 2. A number of action units (AU7 –lid tightener, AU9 –nose wrinkle, AU45 –blink) are not present in any of the images and hence do not covary with the projection. Thus, both commonalities and idiosyncrasies in facial behaviours were observed over subjects.

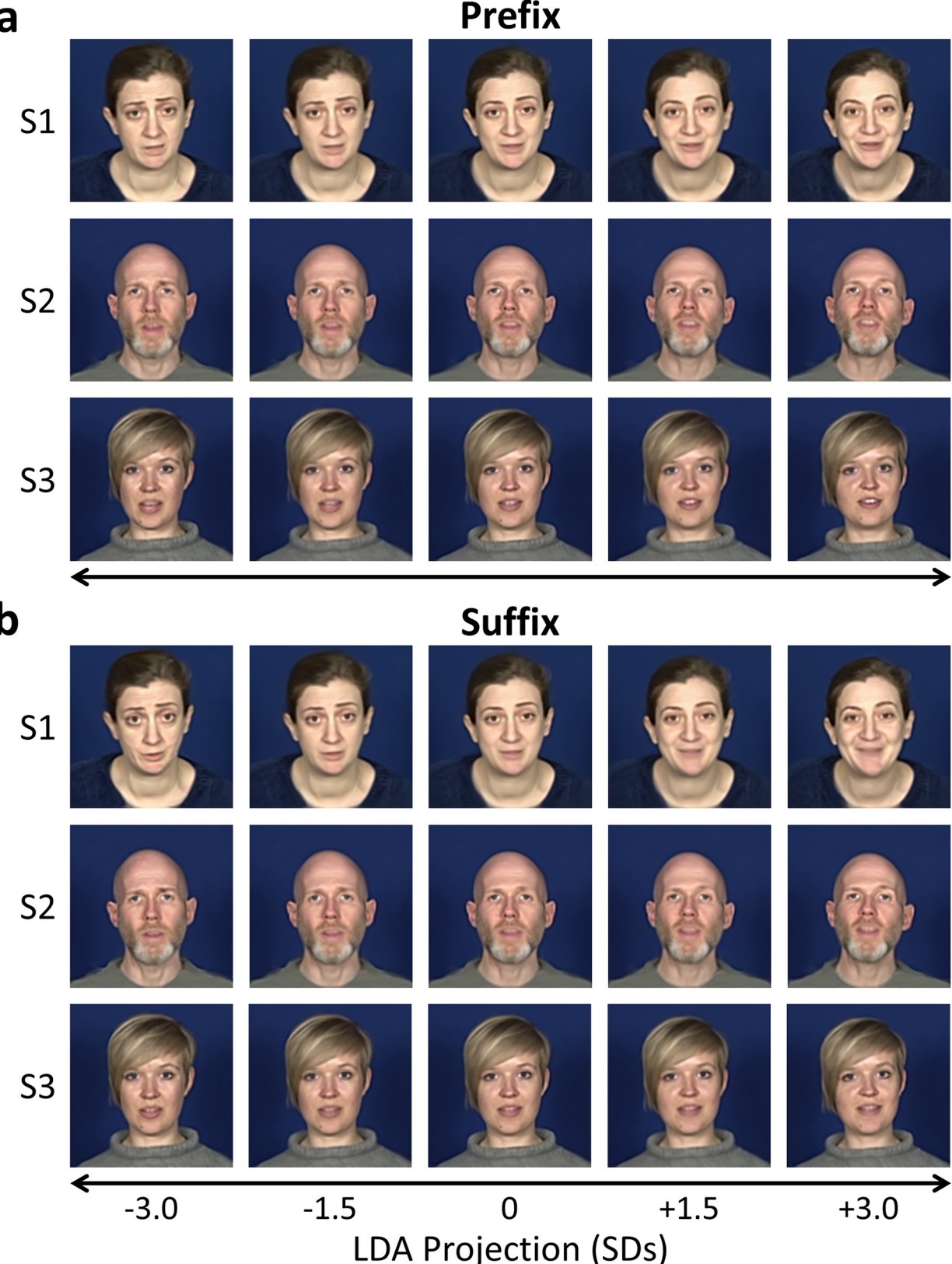

**Fig 4. Projections along LDA discriminant dimension.** Samples were identified along the discriminant dimension and back-projected into the image space for (a) prefix and (b) suffix clips. Examples at 0, ±1.5, and ±3 standard deviations along the dimension are illustrated for each subject.

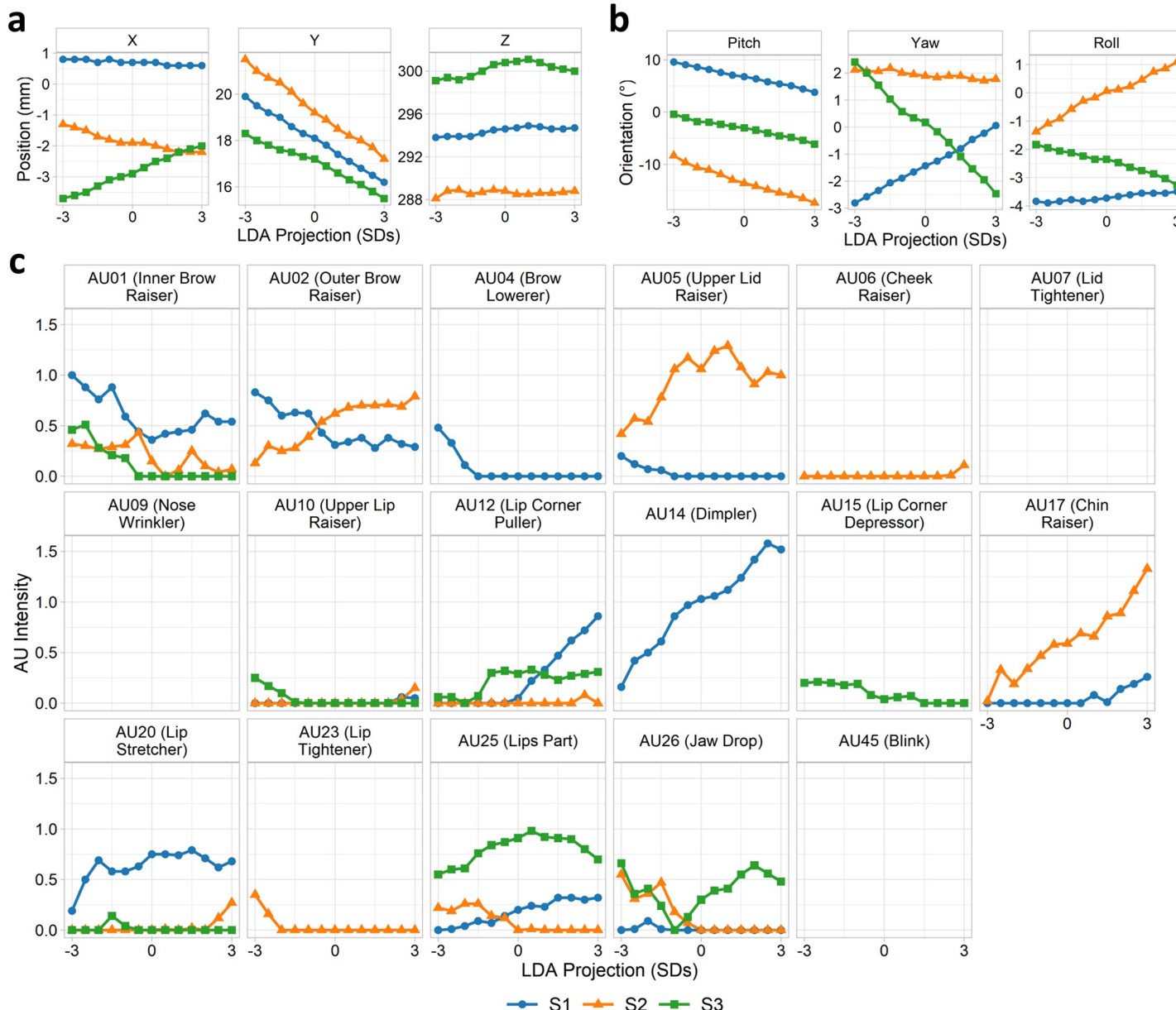

**Fig 5. Prefix clips: FACS features of McGM-PCA LDA projections.** FACS features extracted for images projected along LDA discriminant dimension, including (a) head position, (b) head orientation, and (c) action unit intensities. Points are omitted from graphs where the action unit was not detected in any image.

Finally, we sought to quantify the behavioural relevance of the discriminant dimension. Five naive human observers rated the back-projected images on their emotional valence on a scale ranging from -1 (negative) to +1 (positive). Responses are illustrated in Fig 7. There was a strong positive correlation between observers' ratings and position along the discriminant dimension (mean Pearson's $r$ = .83 ± .05 SEM; all $p$ < .01 except for Rater 4's ratings of Subject 2 where correlations were not significant). There was also high inter-rater reliability, with raters' responses all positively correlating with each other (mean Pearson's $r$ = .81 ± .02 SEM; all $p$ < .001). Thus, the images derived from this data-driven procedure convey behaviourally interpretable dimensions of facial emotion.

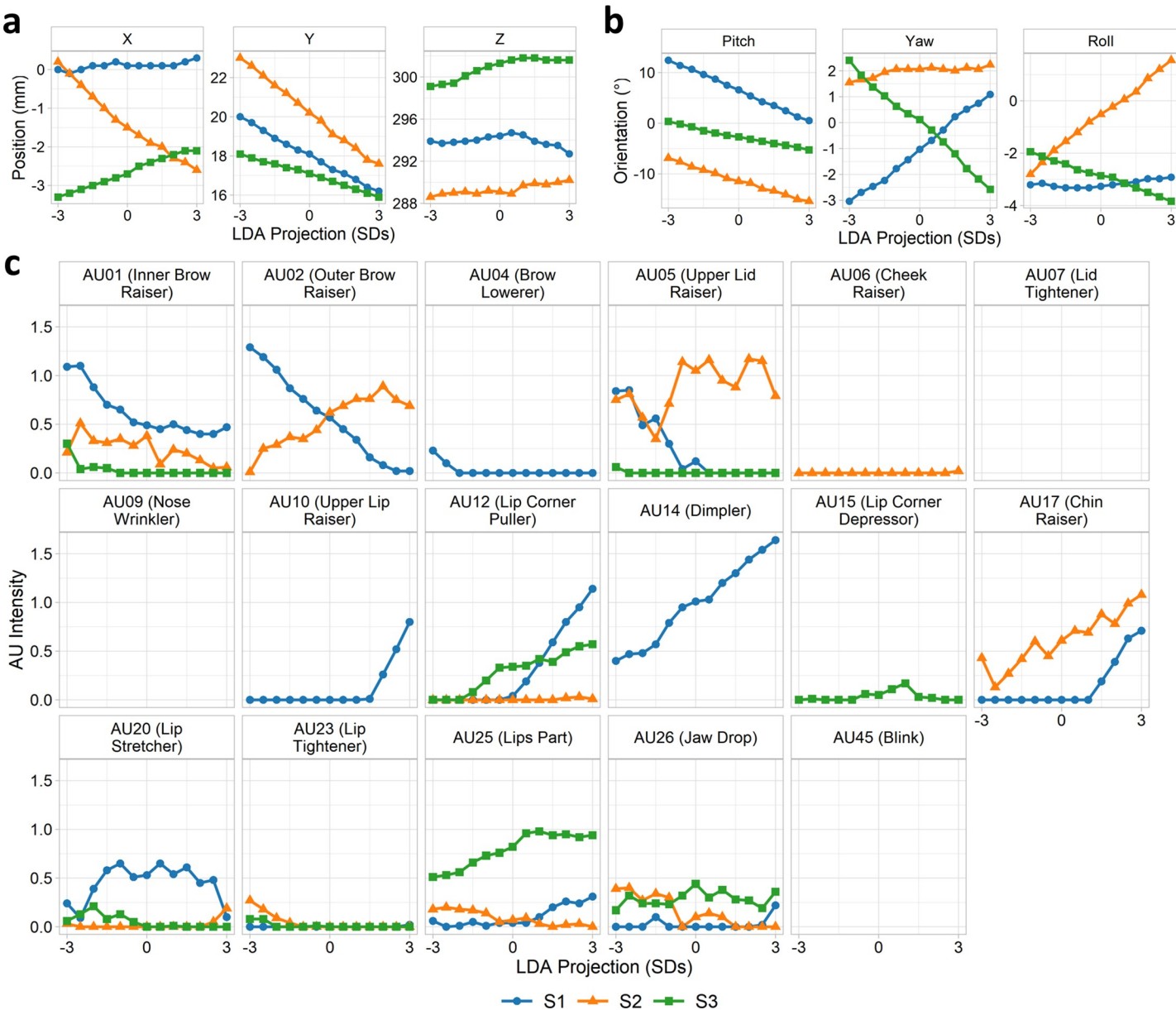

**Fig 6. Suffix clips: FACS features of McGM-PCA LDA projections.** FACS features extracted for images projected along LDA discriminant dimension, including (a) head position, (b) head orientation, and (c) action unit intensities. Points are omitted from graphs where the action unit was not detected in any image.

### Facial action coding analysis

We also tested the ability to discriminate the positive versus negative phrases on the basis of FACS-based features. We used the OpenFace toolbox [26,27] to extract 6 head pose (3D position and orientation) and 17 action unit (AU) intensities for each frame in every clip. We tested three models: one using the AU features alone, one using the head pose features alone, and one using both. We then tested classification accuracy using an LDA classifier within a leave-one-phrase-out cross-validation, in the same manner as for the McGM-PCA model.

The resulting confusion matrices are shown in Fig 2 and the classification accuracies in Fig 3. In all subjects and for both *prefix* and *suffix* clips, the positive versus negative phrases could

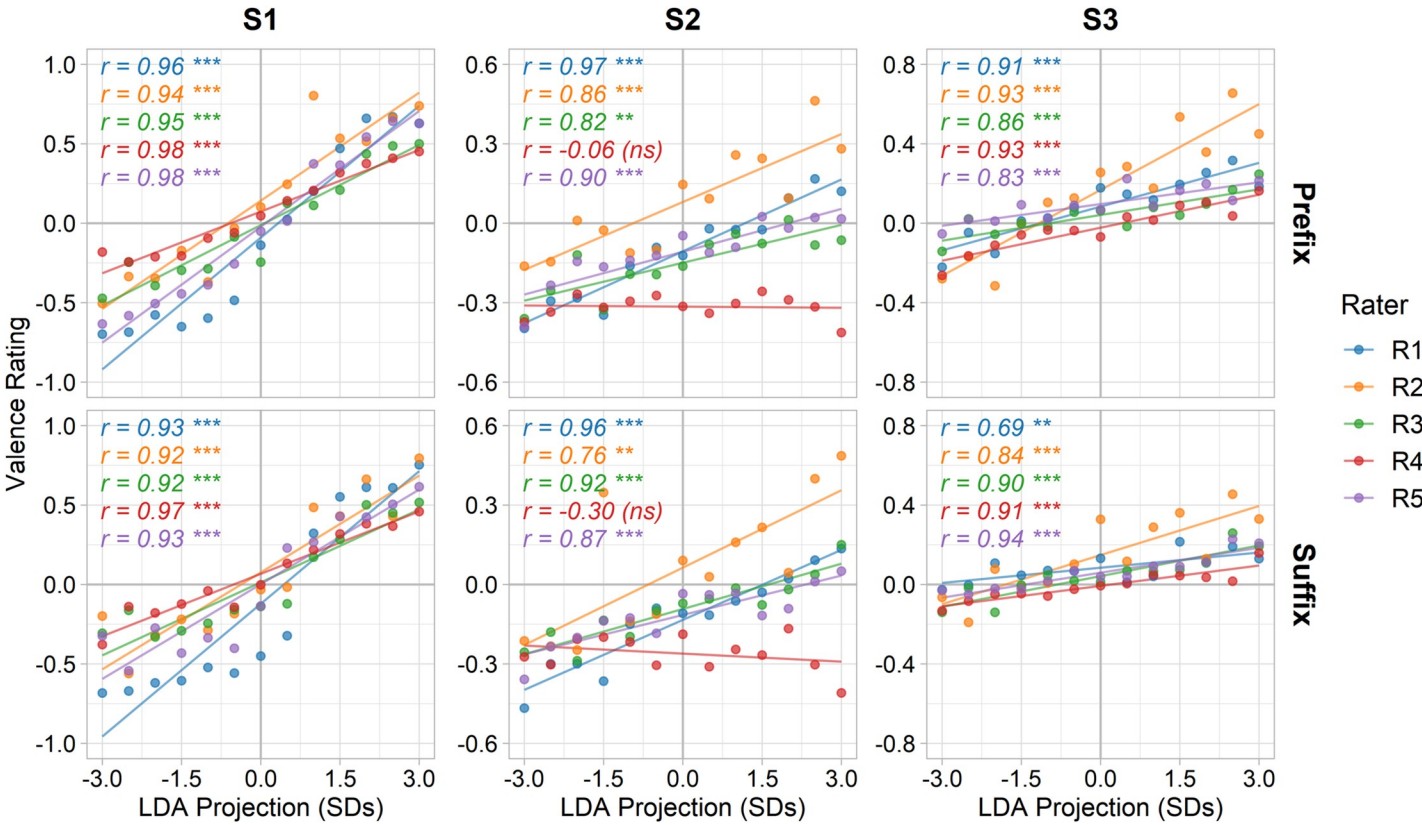

**Fig 7. Behavioural valence ratings along LDA discriminant dimension.** Observers rated images along the discriminant dimension for emotional valence on a continuous scale between -1 (negative) and +1 (positive). Recording subjects are displayed across columns, prefix and suffix clips across rows, and rating subjects by colour. *** p < .001, ** p < .01, * p < .05, (ns) p > .05.

be discriminated well above chance (one-sample t-tests: all $p < .01$). Classification was generally better for the combined AUs+Pose model than for either feature set alone. Nevertheless, classifications appeared diminished relative to the performance of the McGM-PCA model. A series of one-way ANOVAs (Table 1) revealed a significant main effect of model type for all subjects and for both *prefix* and *suffix* clips (all $p < .001$). Planned contrasts (Table 2) revealed that the McGM-PCA outperformed all of the FACS-based models in all subjects (all $p < .05$), except for the comparison against the AUs+Pose model for *suffix* clips in Subject 1 ($p = .134$). Full pairwise comparisons between all models are listed in S2 Table; briefly, the McGM-PCA model again outperforms the FACS-based models, the AUs+Pose model generally outperforms the AUs- and Pose-only models, and the relationship between the AUs- and Pose-only models is more variable over subjects. Thus, while the coarse-scale information provided by the FACS features was sufficient to discriminate the emotional valence, these models were outperformed by the finer-grained pixel-level detail of facial texture and shape changes offered by the McGM-PCA model.

## Cross-subject generalisation

Inspection of the patterns of facial variation revealed by the discriminant dimension (Fig 4) indicates some commonalities across subjects. This suggests that classification of emotional valence may show some degree of generalisation across subjects. To investigate this possibility, we repeated our classification analyses using a cross-subject cross-validation scheme.

**Table 1. ANOVAs testing the main effect of model type on classification performance.**

| Subject | Clip type | F | df | p | $\eta^2$ | $\eta^2_G$ |
|---|---|---|---|---|---|---|
| S1 | Prefix | 190.81 | 1.36, 12.26 | < .001 | .95 | .93 |
| | Suffix | 66.11 | 1.37, 12.37 | < .001 | .88 | .81 |
| S2 | Prefix | 131.82 | 1.66, 14.97 | < .001 | .94 | .87 |
| | Suffix | 61.01 | 1.65, 14.87 | < .001 | .87 | .79 |
| S3 | Prefix | 51.36 | 1.71, 15.42 | < .001 | .85 | .79 |
| | Suffix | 20.30 | 1.22, 10.97 | < .001 | .69 | .59 |
| Cross-subject | Prefix | 28.81 | 2.21, 63.97 | < .001 | .50 | .41 |
| | Suffix | 29.83 | 2.56, 74.31 | < .001 | .51 | .40 |

There is unlikely to be a one-for-one correspondence in the dimensions of the feature spaces between subjects, so the data must first be aligned across subjects. We adapted a hyperalignment procedure [28] which applies a Procrustes transformation to align the data using translations, rotations (including reflections), and a global scale factor. To mitigate overfitting, the data used for calculating the alignment must remain independent of the data used for testing classification accuracy. To this end, we employed a nested cross-validation scheme. First, an outer (5-fold) leave-two-phrases-out cross-validation scheme was used to perform the hyperalignment. Datasets were aligned across subjects based on all samples in 8 out of the 10 phrases in each class, then samples in the remaining 2 phrases in each class were brought into the aligned space using the transformation parameters identified from the outer-training set. Next, within just the outer-test set, LDA classification accuracy was assessed via an inner (6-fold) cross-validation using a simultaneous leave-one-subject- and leave-one-phrase-out scheme. The LDA algorithm was fit to all samples in 2 of the 3 subjects for one of the phrases in each class, then tested on all samples for the third subject on the other phrase in each class. This procedure was repeated across all folds of the inner and outer cross-validation schemes, yielding 30 folds in total.

The resulting confusion matrices are shown in Fig 8, and a summary of the classification accuracies are shown in Fig 3. Classification performance is clearly reduced relative to the within-subject analyses, indicating that some idiosyncratic information is lost. Nevertheless,

**Table 2. Planned contrasts of McGM-PCA model classification accuracies against the FACS-based models.**

| Subject | Baseline model | *Prefix* clips | | | *Suffix* clips | | |
|---|---|---|---|---|---|---|---|
| | | t | df | p | t | df | p |
| S1 | AUs | 6.34 | 27 | < .001 | 2.86 | 27 | .023 |
| | Pose | 22.46 | 27 | < .001 | 12.87 | 27 | < .001 |
| | AUs + Pose | 4.71 | 27 | < .001 | 2.02 | 27 | .134 |
| S2 | AUs | 18.88 | 27 | < .001 | 12.89 | 27 | < .001 |
| | Pose | 14.84 | 27 | < .001 | 9.94 | 27 | < .001 |
| | AUs + Pose | 11.51 | 27 | < .001 | 8.18 | 27 | < .001 |
| S3 | AUs | 11.49 | 27 | < .001 | 7.56 | 27 | < .001 |
| | Pose | 9.73 | 27 | < .001 | 5.17 | 27 | < .001 |
| | AUs + Pose | 6.27 | 27 | < .001 | 3.38 | 27 | .006 |
| Cross-subject | AUs | 5.06 | 87 | < .001 | 7.52 | 87 | < .001 |
| | Pose | 8.52 | 87 | < .001 | 8.28 | 87 | < .001 |
| | AUs + Pose | 7.47 | 87 | < .001 | 7.22 | 87 | < .001 |

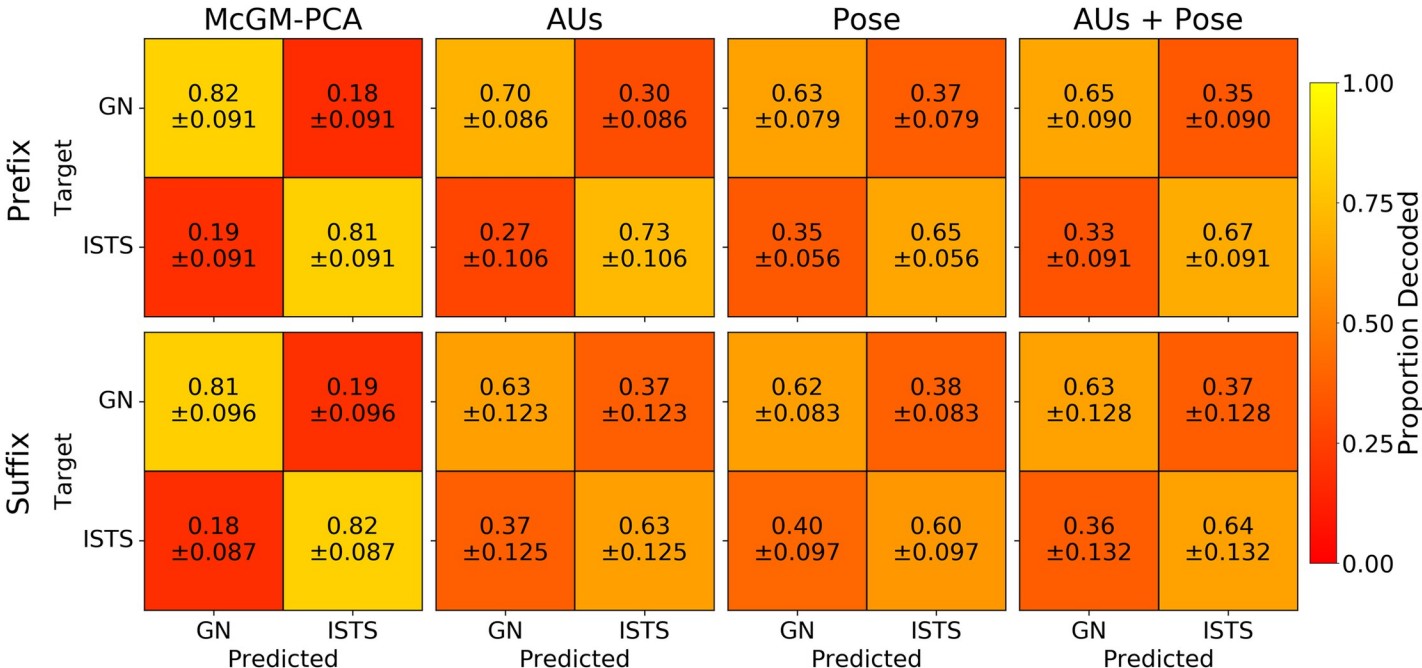

**Fig 8. Cross-subject cross-validated confusion matrices.** Matrices indicate mean proportion of LDA classifications for each combination of "good news" (GN) and "I'm sorry to say" (ISTS) classes. Models are displayed across columns, and top and bottom rows display results for prefix and suffix clips respectively. Annotations indicate means and standard deviations of values over cross-validation folds.

classification remains well above chance for all models for both *prefix* and *suffix* clips (one-sample t-tests: all $p < .001$), indicating some generalisation of facial variation across subjects too. Consistent with the within-subject analyses, one-way ANOVAs revealed a significant main effect of model type for both *prefix* and *suffix* clips (Table 1; all $p < .001$), and planned contrasts showed the McGM-PCA model outperformed all the FACS-based models (Table 2; all $p < .001$). Full pairwise comparisons between all models are listed in *S2 Table*; briefly, the McGM-PCA model again outperforms the FACS-based models, the AUs-only model outperforms the Pose-only model for *prefix* clips only, and no other comparisons reach significance.

## Discussion

In this study we developed a novel protocol for eliciting natural and dynamic emotional facial expressions, without requiring subjects to deliberately or artificially pose specific expressions. Image registration using a multiple-channel gradient model combined with a principal components analysis was able to capture the critical dimensions of facial variance, providing a data-driven solution to identifying emotional facial cues from the natural facial behaviours. A classification algorithm (linear discriminant analysis) was then not only able to discriminate the facial emotions, but also extract a behaviourally relevant dimension of emotional valence. Projections along this dimension were consistently perceived as varying in valence by human observers, demonstrating the psychological relevance of this computational approach. Both classification performance and projections along the discriminant dimension appeared highly similar between *prefix* clips (which contain a common linguistic component) and *suffix* clips (which do not), indicating the features underlying the discrimination of facial emotion were independent of linguistic cues. Indeed, back-projections along the discriminant dimension for each clip type produced highly similar visualisations (Fig 4), indicating substantial overlap in

the model representations of the *prefix* and *suffix* clips. The discrimination of facial emotion was also observed to generalise across subjects, demonstrating a degree of commonality in patterns of facial variation across people.

We derived emotional expressions from natural and dynamic facial behaviours. This protocol contrasts substantially with more traditional paradigms that require actors to deliberately pose specific expressions under tightly controlled experimental conditions. A limitation of such approaches is that they risk controlling away potentially important sources of facial variation, such as within-person variability [5]. By back-projecting samples along the discriminant dimension into the image space, we could visualise the patterns of facial variation underpinning the discrimination of emotional valence. Some of these patterns included changes consistent with those seen in traditional face stimuli, such as variations in the internal features of the face that would be adequately described by facial action units [6,24,25]. However, we also identified patterns of behaviour that, while informative, would nevertheless typically be controlled away in traditional stimuli. For instance, one of the most prominent and consistent changes was seen in the head posture—characterised by a dipping of the head when conveying messages with negative valence and a raising of the chin when conveying messages with positive valence—yet head position is often rigidly controlled in traditional face stimuli. Interestingly, the relevance of head posture has previously been discussed in relation to expressions falling outside of the basic universal emotions included in traditional stimuli. For instance, a lowering of the head has been associated with emotions such as shame and embarrassment, while raising the head has been linked to expressions of pride [29–31]. Previous studies have identified the importance of using so-called *ambient* face images, which maintain natural patterns of facial variation, for tasks such as face identification [5] and perception of social traits [14]. Our results therefore further highlight the importance of such natural facial behaviours in the perception of emotional expression. Such a proposal is consistent with recent discussions of the role of facial variability within emotion categories in supporting initial learning of those categories [32].

The data-driven approach employed here is similar to previous studies [17,33] demonstrating that linear discriminant analysis applied to a PCA-based face space can discriminate emotion categories. However, such studies still required traditional controlled face stimuli as inputs to the analysis. Here, we extend this by demonstrating that data-driven approaches can also be applied to ambient and dynamic faces that embody more natural variability. These data-driven approaches help overcome limitations posed by experimenter bias in the selection of expressions that can potentially confound more traditional face stimuli. The combined PCA and linear discriminant analysis approach described here has also been applied to the classification of other facial properties such as identity, sex, and race–using both controlled [33] and natural [34] face stimuli. This demonstrates the approach's utility for the extraction of multiple facial properties beyond emotion. It should be noted that there are also several alternative computational approaches to extracting facial features. Notably, deep neural networks have demonstrated high performance in recognising facial identity [35,36] and expressions [37,38]. Nevertheless, our model was able to discriminate facial expressions with a high degree of accuracy, and the simplicity of this approach compared to others offers its own advantages such as greater transparency of the model representations.

We also compared the McGM-PCA model to models based on more traditional facial coding metrics, including estimates of head pose and facial action units [24,25]. These models were also able to discriminate the emotional valence of the faces, but were consistently outperformed by the McGM-PCA model. This is likely due to the relatively coarse level of detail that the facial action coding models provide, compared to the fine-scale pixel-level detail in facial texture and shape change offered by the McGM-PCA model. The use of PCA to capture critical dimensions of facial variation from images has previously been employed to describe facial

emotions [17,39], but also features such as facial identity [23,40,41], gender [42], and race [43]. Such approaches typically apply PCA to the original images or after first morphing the faces to a common template to provide a "shape-free" texture representation [17,41]. Applying factor analysis to facial action units, rather than the images themselves, has also been used to characterise a number of meaningful facial behaviours [39]. Here, we first used the McGM to warp images to a common template, and performed principal components decomposition on the combination of the warped textures and warp components. Warping images to a common template essentially provides a "shape-free" estimate of the visual textures, similar to previous approaches, but by also including the warp components we additionally represent information about changes in facial configuration across frames. This approach is therefore well suited to capturing the critical patterns of variation observed in dynamic faces [20,22].

Sampling along the discriminant dimension of the McGM-PCA model yielded images that varied consistently along a dimension of emotional valence. Human observers reliably perceived the modulation of valence in line with this dimension, demonstrating our data-driven technique was able to extract behaviourally relevant features. These behavioural ratings also highlighted individual differences in facial behaviours; for instance, observers typically provided more extreme ratings for Subject 1 than the other subjects. Indeed, the back-projected images (Fig 4) do subjectively appear to show a greater range of expression for Subject 1. Similarly, analysis of the head poses and facial action units in the projected images also revealed both commonalities and idiosyncrasies in facial behaviours over subjects. Interestingly, while both behavioural ratings and facial action coding metrics indicate individual differences, human observers nevertheless perceived a consistent direction of valence along the discriminant dimension, while the coding metrics frequently showed opposing directions of effect between subjects. This suggests a potential disconnect between facial action coding metrics and human perceptions of facial emotion when dealing with more natural patterns of facial behaviour.

The modulation and perception of valence along a single continuum is consistent with continuous models of facial emotion [10,11] that propose representing faces in multi-dimensional face spaces defined by multiple emotional dimensions. By contrast, categorical models of face perception posit that emotions are organised into a number of discrete classes, with particular focus given to universal basic emotions [6]. Such a view is less consistent with our description of a continuous valence dimension, though the nature of this representation may be somewhat task dependent. For instance, our behavioural experiment may have yielded more categorical responses if we had employed a categorisation rather than a rating task [12]. Here we applied our model to a dimension of emotional valence, but it could in principle be extended to other emotional dimensions. For instance, continuous models often identify *arousal* as another key emotional dimension [10,11]. Future research may further consider interactions between valence and other emotional dimensions; for instance, positive versus negative valence combined with high arousal could yield surprised expressions but with excited versus tense interpretations. The technique could also be extended to further facial dimensions beyond emotion, for instance to dimensions of social evaluation such as dominance, trustworthiness, and perceived political affiliation [2,14,44].

The classification of emotional valence also generalised across subjects. A clear decrement in performance was observed relative to the within-subject analyses, but discrimination remained well above chance. Thus, while patterns of natural facial variation embody some idiosyncratic features that do not generalise across subjects, they also include some common features that do generalise. This result is particularly striking given the relatively small number of subjects that were recorded here, suggesting that such features are highly consistent across individuals and do not require larger populations to be identified. However, this finding does

contrast with recent research suggesting that PCA face-spaces are relatively idiosyncratic and should not generalise well across people [45]. One possible explanation is that whereas the changes in the mouth, say, in a smile might be idiosyncratic and differ across individuals and principal components, the key feature picked up in the present analysis is a global postural change—chin tucking against chin raising. This global transformation may be more universal and will generalise even if featural changes may differ. Cross-subject generalisation is consistent with similar approaches that have previously demonstrated generalisation of the McGM-PCA model over facial genders [21]. In this case, if a male face is coded as longer and thinner than average, and this transformation is projected into a female face space, then an apparent sibling-like similarity across the genders can be generated.

In conclusion, we advance a novel protocol for eliciting natural patterns of facial behaviour from dynamic faces. A data-driven method was able to both discriminate the emotional state of the faces and recover behaviourally relevant emotional dimensions. This method reproduced patterns of facial variance frequently seen in traditional face stimuli (such as changes in internal facial features), but also revealed dimensions that would typically be omitted from such stimuli (such as a dipping versus raising of the head posture between negative and positive valences). Here we applied this model to a dimension of emotional valence, but this procedure could be readily extended to other emotional and non-emotional facial dimensions.

## Methods

### Recordings

Three subjects (2 females, 1 male, age range 26–42) were video recorded. The study was approved by the ethics committee of the School of Psychology at the University of Nottingham (ethics approval number: 717) and conducted in accordance with the guidelines and regulations of this committee and the Declaration of Helsinki. All subjects provided informed written consent to take part in the study and for their likeness to be used in publication.

Recordings were made in an anechoic chamber against a uniform visual background. Videos were acquired on a Sony HXR-NX5U NXCAM camera connected to an Atomos Ninja-2 recorder that recorded videos in Apple ProRes RAW format. Videos were acquired at a resolution of 1920x1080 pixels and at 25 fps with a 6.67ms exposure. Videos were then encoded using MPEG-4 lossless compression prior to further processing.

Subjects each delivered a total of 20 unique phrases to the camera each conveying either positive or negative news (10 phrases of each). A list of the phrases is provided in *S1 Table*. Subjects 1 and 2 performed 15 repeats of each phrase (300 phrases total), and Subject 3 performed 16 repeats (320 phrases total). Subjects were instructed to deliver the phrases in whatever manner felt most natural to them; they were not instructed to deliberately pose or perform any specific expressions or actions. To aid subjects in delivering their phrases in a natural way they viewed pre-recorded silent videos of a variety of recipients, presented on a teleprompter directly in front of the camera. Recipient videos were obtained from YouTube and depicted video-conference style calls, helping give subjects the impression of delivering their phrase to a person listening to them.

For each phrase, we generated two types of clips. *Prefix* clips comprised the first 1.44 seconds (36 frames) after the onset of each phrase, primarily including the initial "Good news" / "I'm sorry to say" portion of each phrase and a small part of the later sentence. The onset of each phrase was identified in a semi-automated procedure by identifying the increase in audio amplitude from a spectrogram of the audio signal, and then applying manual corrections where necessary. To rule out potential confounds from the common linguistic information shared across the *prefix* clips, we also generated *suffix* clips that removed the initial common

prefix sections of each phrase and retained only the later sentence portion. Onsets and offsets of the suffix portions were identified in a semi-automated fashion using the Google Cloud Speech-to-Text algorithm (https://cloud.google.com/speech-to-text) to generate timestamps for each word, and then applying manual corrections as necessary. Unlike the *prefix* clips, the *suffix* clips were variable in length.

## McGM-PCA model

A Multi-channel Gradient Model [18–20] (McGM) was used to capture textures and register all images in a sequence to a reference image. First, for each clip a Haar cascade face-detection algorithm implemented in OpenCV (https://opencv.org/) was used to determine the average position of the face within the scene, and the video was cropped to a square bounding box centred on this position and allowing a small border around the face. This helped ensure the face was placed approximately centrally within the scene. These clips were then down-sampled to a resolution of 128x128 pixels using an anti-aliasing filter.

The cropped and downsampled clips were then entered into the McGM. For each frame, a warp vector field was calculated that registered that frame to a standard reference image. The reference was initially set as an individual frame taken from one of the recording sequences. However, to provide a more standardised reference, the original reference was then replaced with the average of all textures after warping. This process was then repeated three times, re-calculating the warps and updating the reference with the average warped texture each time, to allow the reference to stabilise. For each input frame, the McGM produced a 5-channel image comprising the x- and y-direction warp components and a "shape-free" version of the RGB textures warped to the final average reference. Flattening these images to vectors yields an 81,920-dimensional ($128 \times 128 \times 5$) feature space, with each frame in each clip represented as an individual sample within this space. Note that the two frame version of the McGM used here for the purpose of image registration [20] differs from some previous applications of the model that instead computed local image velocities over extended temporal sequences [18,19].

The dimensionality of this feature space was reduced via a principal components analysis (PCA) [17,23], retaining a sufficient number of components to explain 90% of the variance across samples. For the *prefix* clips, applying this to the full dataset yielded 321, 149, and 383 components for each subject respectively. For the *suffix* clips, applying this to the full dataset yielded 346, 135, and 395 components for each subject respectively. For the cross-validated analyses the PCA procedure was modified to ensure independence of training and test splits. For a given cross-validation fold, the PCA was computed based on the samples included in the training set only, and the resulting transformation coefficients were then applied to both the training and test sets. In this way, both datasets are brought into the same PCA-space, but the definition of this space remains independent of the test data. Transformation of the test data into this space will therefore only be appropriate if the principal components generalise between training and test sets. For the leave-one-phrase-out cross-validation utilised for within-subject analyses, this yielded a median (across folds) 310, 146, and 373 components for *prefix* clips, and 335, 133, and 378 components for *suffix* clips for each subject respectively. For the leave-two-phrases-out outer cross-validation employed for the cross-subject analyses, a common number of components were retained across subjects that explained a minimum of 90% variance in all subjects. This yielded a median (across folds) 350 and 271 components for the *prefix* and *suffix* clips respectively.

## Facial action coding models

We used the OpenFace toolbox (v2.2.0; https://github.com/TadasBaltrusaitis/OpenFace) [26,27] to automatically extract a number of facial features from the clips following the Facial

Action Coding Scheme (FACS) [24,25]. Specifically, for each frame in each clip we extracted 6 head pose features (3D position and orientation parameters) and 17 action unit (AU) intensities (AUs 1, 2, 4, 5, 6, 7, 9, 10, 12, 14, 15, 17, 20, 23, 25, 26, and 45). Prior to processing, each clip was cropped to a central square but left at full resolution (1080p). We constructed three models based on: the AU intensities alone, the head pose alone, and the AU intensities and head pose information combined. Each frame in each clip is therefore represented as a sample in a 17-, 6-, or 23-dimensional feature space for the AUs-only, Pose-only, and AUs+Pose models respectively. No dimensionality reduction was required as these spaces are already relatively low-dimensional.

## Linear discriminant analysis

We employed a Linear Discriminant Analysis (LDA) classification algorithm to discriminate the positive versus negative emotional valences of the clips based on their representations by the McGM-PCA and FACS-based models. We tested the cross-validated classification accuracy of each model, and also back-projected images from along the discriminant dimension of the McGM-PCA space.

**Within-subject classification.**   We tested the within-subject classification accuracy using a (10-fold) leave-one-phrase-out cross-validation scheme. In the case of the McGM, PCA decomposition was also applied within this cross-validation scheme (see above). For each subject and each model, datasets were partitioned into a training set comprising all samples in all clips for 9 of the 10 phrases in each class, and a test set comprising all samples in all clips for the remaining phrase in each class. In each set independently, data were normalised by z-scoring along each feature dimension. The LDA algorithm was then fit to the training set, and predicted class labels were generated for the test set. Classification accuracy is given as the proportion of predicted class labels that match the target labels within the test set. This procedure was then repeated for the remaining folds of the cross-validation, and accuracies were averaged over the folds.

**Cross-subject classification.**   The cross-subject analyses further require that subjects' feature spaces be aligned together as there will not necessarily be a one-to-one correspondence in features across subjects. We adapted a hyperalignment [28] procedure, which aligns the data using an affine Procrustes transform allowing for translations, rotations (including reflections), and a global scale factor. This requires a common number of samples across subjects, so the final 20 clips in Subject 3 were discarded such that all subjects had 300 clips, and then all clips in all subjects were truncated to the length of the shortest clip in any subject such that all clips comprised the same number of frames. The hyperalignment itself is performed in a two-stage process. The first stage aligns the second subject's data to the first, then aligns the third subject's data to the average of the preceding two and updates the average accordingly. In the second stage, the alignments for each subject are recomputed to point to the final average obtained from the first stage. Applying these transforms will bring all subjects' data into alignment in a common group feature space.

To mitigate overfitting, both the hyperalignment and classification analyses were cross-validated using a nested cross-validation scheme. The hyperalignment was performed within an outer (5-fold) leave-two-phrases-out cross-validation scheme. In the case of the McGM, the PCA transformation was also performed within this cross-validation (see above). As the alignment procedure requires a common number of dimensions over subjects, a common number of principal components were retained over subjects such that a minimum of 90% of the variance was explained in all subjects. PCA coefficients (McGM only) and Procrustes parameters for each subject were calculated based on all samples in all clips for 8 of the 10 phrases in each

class. The PCA decomposition (McGM only) and Procrustes transforms were then applied to all samples in all clips for the two held-out phrases in each class. Within the outer-test set only, LDA classification was then performed within an inner (6-fold) simultaneous leave-one-subject- and leave-one-phrase-out cross-validation. Samples were normalised by z-scoring along each feature within inner-training and inner-test sets independently. The LDA classifier was fit to all samples in all clips for one of the phrases in each class for 2 of the 3 subjects. Predicted class labels were then generated for all samples in all clips for the other phrase of each class in the held-out subject. As per the within-subject analyses, classifier accuracy was assessed by comparing predicted and target class labels. Successful classification thus depends on generalisation across both subjects and phrases. Repeating this process across all folds of the inner and outer cross-validations yielded 30 folds in total, and classification accuracies were averaged over folds.

**Analysis of model performance.** All statistical analyses of models were performed within each subject and for the cross-subject analyses independently. Classifier accuracies were tested against chance (50%) via one-sample t-tests, subject to a Holm-Bonferroni correction [46] for multiple comparisons across models. Comparisons between models were made by one-way ANOVAs, entering the model type as a repeated-measures factor (with levels: McGM-PCA, AUs-only, Pose-only, and AUs+Pose). A Greenhouse-Geisser sphericity correction was applied to all tests. Effect sizes are reported in units of eta-squared and generalised eta-squared [47,48]. To further interrogate differences between models a series of planned contrasts were run testing each of the FACS-based models against the McGM-PCA model subject to a Dunnett correction for multiple comparisons [49]. For completeness, we also performed the full set of pairwise comparisons between all models (see *S2 Table*) subject to a Tukey correction for multiple comparisons [50]. All tests employed an alpha criterion of 0.05 for determining significance.

**Projection along discriminant dimension.** To identify the facial features underlying discrimination of emotional valence, we additionally projected samples along the within-subject discriminant dimensions identified within the McGM-PCA spaces. As this does not require cross-validation, we re-fit the LDA algorithm to the full dataset (including all 10 phrases in each class). Again, data were normalised by z-scoring along each feature prior to classification. The discriminant dimension was identified as the line orthogonal to the decision boundary and which passes through the centroids of both classes. We quantified distance along this line by taking the projection of all data samples onto the discriminant dimension then measuring their variance along it. We then generated a series of samples ranging between ±3 standard deviations along the dimension and back-projected them to the image space for visualisation. For each sample, we reversed the z-scoring operation to return to the McGM-PCA space, inverted the PCA to return to the McGM space, then inverted the warp components to unwarp the image back to the image space. To aid visualisation, the visual contrast of the images was enhanced via unsharp masking.

We extracted FACS-based features for the projected images using the OpenFace toolbox [26,27]. For each subject and for both *prefix* and *suffix* clips, we generated images for 13 samples evenly spaced between ±3 standard deviations along the discriminant dimension in 0.5 standard deviation steps. We extracted 6 head pose features (3D position and orientation) and 17 action unit intensity features–see methods on facial action coding models for full details. For each of the 23 features, we then plotted the feature intensities against the position along the discriminant dimension (Figs 5 and 6).

## Behavioural experiment

We conducted a behavioural experiment to determine human perception of the emotional valence of images back-projected from the McGM-PCA LDA discriminant dimension. Five

participants took part in the experiment (3 females, 2 males, age range 23–35). The study was approved by the ethics committee of the School of Psychology at the University of Nottingham (ethics approval number: F1249) and conducted in accordance with the guidelines and regulations of this committee and the Declaration of Helsinki. Participants provided informed consent via an electronic form before participating in the study. To distinguish from the participants used in the original video recordings, participants in this behavioural experiment are referred to as *raters*.

For each of the three recording participants and for both *prefix* and *suffix* clips, we generated 13 images between ±3 standard deviations along the discriminant dimension in 0.5 standard deviation steps, yielding 78 images total. The experiment employed a block design, with each of the images presented once in every block in a randomised order. Each image was displayed for 1 second, followed by a sliding scale marked with "negative", "neutral", and "positive" at the ends and midpoint. Raters indicated their perception of the emotional valence of the preceding face by clicking along the slider. Raters were allowed unlimited time to enter their response. Each rater completed 4 blocks in total. Raters were not provided with any feedback or other cues as to how their ratings compared to the position along the discriminant dimension. The experiment was run online using PsychoPy3 and Pavlovia ([https://pavlovia.org/](https://pavlovia.org/)) [51].

Ratings along the scale were coded by a numerical range between -1 (most negative valence), 0 (neutral), and +1 (most positive valence). For each rater, responses were averaged over the 4 blocks, ultimately yielding one sample per image. Ratings were then correlated against the discriminant dimension position (in standard deviations) for each clip type and each recording participant independently. We also measured inter-rater reliability by concatenating each rater's responses over recording subjects and then correlating between raters. A Holm-Bonferroni correction for multiple comparisons [46] was applied over raters. An alpha criterion of 0.05 was used for determining significance.

## Supporting information

**S1 Table. List of the ten positive and negative phrases delivered by each subject.**
(DOCX)

**S2 Table. Full post-hoc contrasts of classification accuracies between models.** Significant contrasts are highlighted in bold.
(DOCX)

**S1 Video. McGM-PCA LDA projection for Subject 1, prefix clips.**
(AVI)

**S2 Video. McGM-PCA LDA projection for Subject 1, suffix clips.**
(AVI)

**S3 Video. McGM-PCA LDA projection for Subject 2, prefix clips.**
(AVI)

**S4 Video. McGM-PCA LDA projection for Subject 2, suffix clips.**
(AVI)

**S5 Video. McGM-PCA LDA projection for Subject 3, prefix clips.**
(AVI)

**S6 Video. McGM-PCA LDA projection for Subject 3, suffix clips.**
(AVI)

## Author Contributions

**Conceptualization:** David M. Watson, Ben B. Brown, Alan Johnston.

**Data curation:** David M. Watson, Ben B. Brown, Alan Johnston.

**Formal analysis:** David M. Watson, Ben B. Brown, Alan Johnston.

**Funding acquisition:** Alan Johnston.

**Supervision:** Alan Johnston.

**Writing – original draft:** David M. Watson.

**Writing – review & editing:** David M. Watson, Ben B. Brown, Alan Johnston.

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
