## [Decision Letter · Decision Letter 0]

15 Jul 2020

Dear Dr. Watson,

Thank you for your patience in awaiting our response to your manuscript "A data-driven approach to extracting naturalistic behaviours from dynamic faces." We have now received reviews from three experts in the area you are investigating. I thank them for their time and effort in evaluating this manuscript. 

As you will read below, the reviewers shared enthusiasm for the work but also raised a number of queries that advise against publication of the manuscript in its current form. Each reviewer has provided detailed, clear, and constructive descriptions and explanations of each of their points such that they need not be repeated here in further detail. We hope that you will find this feedback useful. Briefly stated, the reviewers converged in identifying areas of the manuscript that require clarification and development, including aspects of the methodological approach and the theoretical grounding and significance of the results. In sum, the reviewers agree that the work has the potential to provide a knowledge contribution, pending these specific revisions. 

On this basis, we invite you to resubmit the manuscript following major revisions to address the reviewers' queries. The revised manuscript will then likely to be sent to the original reviewers for evaluation.

If you decide to revise and resubmit the manuscript, please upload the following:

[1] A letter containing a detailed list of your responses to the reviewers' comments and a description of the changes made to the manuscript. Please note while forming your response, if your article is accepted, you may have the opportunity to make the peer review history publicly available. The record will include editor decision letters (with reviews) and your responses to reviewer comments. If eligible, we will contact you to opt in or out.

Sincerely,

Rachael Jack, PhD

Guest Editor

PLOS Computational Biology

Wolfgang Einhäuser

Deputy Editor

PLOS Computational Biology

Reviewer's Responses to Questions

**Comments to the Authors:**

Reviewer #1: The authors develop a multi-channel gradient motion (McGM)model combined with principal components analysis to capture information about facial variance over frames in moving dynamic faces. This representation of the dynamic information was used with a classifier to predict valence very successfully. The authors also show that by “tracing samples along the discriminant dimension and back projecting into the image space” they could visualize and interpret features that give rise to emotional valance. Notably, both internal features as well as head pose/movements were reliable signals of message valence.

To quantify the relationship between the McGM features and valence, the authors

compared these features to the extracted Action Units (AUs), Pose, and AUs + Pose. The results show that the McGM features supported more accurate valence classification than did the

AU and pose models tested. A simple human behavioral test supported the conclusion that human behavior can be tracked by variations in the information extracted by the McGM model. Finally, the authors show some degree of generalization across subject classification for all models, but again, with better generalization for the McGM model.

Evaluation

The work provides a nice illustration of the use of the McGM model for extracting valence information from a dynamic face that is captured during natural movement. On the positive side, the model is elegant and effective, and it makes use of detailed facial motion information in naturalistic stimuli. The manuscript is well-written and the experiments are well-designed to test the utility of the model. On the less positive side, there were some places in the paper that were not entirely clear on methods. Consequently, there are some parts of the procedure that are either problematic or are not well enough described to be sure they do not confound variables in the test. There are also only weak links to theory here (see below). These concerns are described below, along with a number of minor points.

Major Comments.

1. I was unsure about the unit of comparison between the McGM features and the AU and Pose features. From what I understand, the McGM features that were assigned to the category of “positive” or “negative” were based on the motion of the full video sequence (line 154-155), whereas the AU’s were done frame by frame (222-224). In short, I really could not extract from the text, exactly what was done. I would worry about the comparison of a frame-by-frame method to a whole-video motion method comparison. I wonder if this is fair comparison. AUC’s are most accurate in higher-intensity frames. It seems possible that some of the frames then would contain excellent valence information, but the ones in between would contain weak or no valence information. Therefore, classification may be succeeding well on intense frames, but performance measures would be diluted by interim frames.

2. The work seems very driven by empirical rather than theoretical concerns, though it is not entirely clear what direction the authors want to take. If this is just a “performance paper”, then more should be said about other computational approaches from face tracking work in the computer literature. If there is a more biological element, some argument should be made on neural or psychological grounds. The finding that behavioral performance relates to McGM features is probably not enough. Note that the Discussion mostly restates the results. Some clarity of the higher level motivation of the work would be beneficial.

Minor comments:

1. The title seems a bit off the mark for what is actually done in the study.

2. The use of the word “data-driven” should be explained or defined. It seems to make an implicit contrast to AU’s ….but it is not clear.

3. Also, “facial expressions” is used a bit loosely here. Valence and expressions are not the same thing. And, one can imagine that expressions (e.g., surprise) can be either positive or negative valiance. More care is needed in terminology.

4. Line 238 finer grained detail – or do you mean motion? Both differ between the elements of the comparison.

5. Discussion- How much beyond valence detection do the authors think the method will go? Again, the topic of facial expression begins in the intro, but dynamic facial behaviors are the focus of the discussion.

Reviewer #2: The current manuscript describes a bottom-up approach for capturing the defining features of positive and negative facial expressions from natural occurring displays. The study involves a novel technique to do this and the authors present their findings in a very comprehensive way. The results reveal that the technique is successful in extracting features of facial displays that are recognized by independent (human) raters as conveying positive and/or negative feelings. The authors discuss how the technique could be extended to other displays.

I found this a very interesting paper that I enjoyed reading. I have a few minor comments and some questions for further clarifications that I will describe below, but I did not have any major reservations about the methodology (as far as I understood it) and the interpretation of the findings.

Before giving my specific comments I have to declare that I have insufficient knowledge of Multi-channel Gradient Modelling, the employed method of Discriminant Analysis and Hyperalignment to judge whether these techniques were appropriate and executed correctly. However, I should say that I found the descriptions and explanations of these techniques very clear. In my review I focus on some aspects of these techniques that were less clear to me and that could perhaps benefit from some further explanation, and aspects of the work related to the FACS codes of the extracted images and the ratings involving human observers.

First, I think some more can be said about the FACS codes of the extracted displays. Interestingly, there appears to be very little overlap in the patterns of FACS codes extracted from the different subjects (Figures S1 and S2). The fact that the human raters were able to differentiate the back-projected images in terms of their valence consistently between subjects, despite the fact that there was such disparity between subjects in terms of the morphological features of the displays is something that could be discussed further. The most consistent pattern of Action Units (AUs) associated with positivity/negativity across subjects were AU1, AU12, and AU17. The relevance of these AUs in this context could be discussed further. Finally, with regards to head movement there is in fact literature that suggests that an upward tilt is associated with positive feelings, while a downward tilt is associated with feelings of dejection (Haidt & Keltner, 1999; Tracy & Robins, 2004; Tracy, Robins & Schriber, 2009) and this could be discussed as well.

Somewhat related to this, the cross-subject decoding accuracy is lower than the decoding accuracy for the individual subjects. This relates to the generalizability of the findings and is something that could be elaborated on more thoroughly in the discussion section. The absolute levels of ratings by the observers of the different subjects are also relevant in this context. For example the displays of S1 seem to be more clearly differentiated by raters than the displays of the other subjects. What may be the reason for this?

One potential limitation of the current technique is that, while the extracted displays are based on filmed displays that involve movement, the back-projected images themselves are static. To what extent is movement an important feature for the accurate decoding of the images? Somewhat related to this, to what extent do the raters rely on making an internal comparison between the images that have been presented to them to judge their positivity / negativity?

Some minor comments and questions:

- There seems to be a large overlap between the displays extracted from the prefix and the suffix parts of the sentences. Can the extend of this overlap be tested more formally?

- Table 2 reports the comparison between the McGM-PCA model and the FACS based models based on planned contrasts, but this is not sufficiently justified. Also, there is potentially relevant information in the comparison between the different FACS based models. Can the post hoc comparisons be reported instead?

- It wasn’t clear to me how the classification accuracies for the cross-subject FACS codes were calculated. Can some more information about this be provided?

- This may be a naïve question, but is it not relevant to also look at the variability of the LDA classifications when reporting the classification accuracy? Is there a Standard Deviation or Standard Error that is associated with the Mean proportion of LDA classifications? If so, can this be reported in Figure 2?

- Stratou, Van der Schalk, Hoegen & and Gratch (2017) employed a PCA based methodology on automatically extracted FACS codes of natural occurring expressions to uncover reliable patterns of facial activity that correspond to certain behaviors. It may be relevant to look into this work.

References

Haidt, J., & Keltner, D. (1999). Culture and facial expression: Open-ended methods find more expressions and a gradient of recognition. Cognition and Emotion, 13, 225–266.

Stratou, G., Van Der Schalk, J., Hoegen, R., & Gratch, J. (2017, October). Refactoring facial expressions: An automatic analysis of natural occurring facial expressions in iterative social dilemma. In 2017 Seventh international conference on affective computing and intelligent interaction (ACII) (pp. 427-433). IEEE.

Tracy, J. L., & Robins, R. W. (2004). Show your pride: Evidence for a discrete emotion expression. Psychological Science, 15, 194–197.

Tracy, J. L., Robins, R. W., & Schriber, R. A. (2009). Development of a FACS-verified set of basic and self-conscious emotion expressions. Emotion, 9, 554–559.

Reviewer #3: This is a nice study, showing how a relatively simple model can extract valence data from naturalistic videos. Using such videos is an important: for some years the expression literature had got bogged down in a self-perpetuating circle of 6 basic expressions and their defining AUs. Another positive feature is that the model is invertible, allowing reconstruction of prototypical stimuli for positive and negative valence.

The least convincing aspect of the work is the use of multiple samples from each of only three participants. It is less surprising that a model can extract commonalities within a relatively self-similar set of videos - especially for the prefix sections. It is therefore reassuring that the model produces almost identical results for the much more varied suffix sections. There is also some testing between participants, which produces results well above chance.

I had a number of queries as I read through the first part of the paper, which were answered when I got to methods. Combined with what looks like a full set of code on the osf site, I think the methodology is well-described.

I think it might be helpful to reference previous work using PCA and LDA to classify naturalistic face images for identity: e.g. Dahl CD, Rasch MJ, Bülthoff I, Chen C-C. Integration or separation in the processing of facial properties - a computational view; Kramer RS, Young AW, Day MG, Burton AM. Robust social categorization emerges from learning the identities of very few faces

L165 'then tested on all samples for the held out phrase.' Am I understanding correctly, this should say phrases, one positive and one negative? Again at L484

Fig 4 is quite convincing, but I expect animations would be better: could you put some in supplementaries, or your OSF site?

L202 Doesn't seem quite right to put these in supplementaries and then talk about them in main text. Lack of space? Note to editor, the first one needs to be in the main text - second can go to S. I think it would be worth drawing attention especially to the yaw data (and explain that is a turn of the head) since two of the subjects show strong correlations, in the opposite direction. Illustrates why within subject model works much better than between. I'd be more inclined to put table 1 and 2 in supplementaries; the numbers don't add much.

L498. The iterative, rolling average description confused me, given that there are only three Ps here. I thought it might go round a loop. The description that follows is simpler - the rolling average bit is a more general description for any N. Because we know N=3, it serves to confuse, here. You could say, align the second to the first, then align the third to the average of the first two in a single sentence.

L540 time? Line, I think.

Peter Hancock

**Have all data underlying the figures and results presented in the manuscript been provided?**

Reviewer #1: Yes

Reviewer #2: Yes

Reviewer #3: Yes

PLOS authors have the option to publish the peer review history of their article (what does this mean?). If published, this will include your full peer review and any attached files.

Reviewer #1: No

Reviewer #2: No

Reviewer #3: No
---

## [Decision Letter · Decision Letter 1]

12 Sep 2020

Dear Dr. Watson,

Thank you for your patience in awaiting our response. The review process took longer than usual, in part due to the current pandemic. We have now received the reviews of the three original experts who first evaluated the manuscript. I thank them for the continued dedication to evaluating the work.

As you will read below, all reviewers agreed that their concerns have been satisfactorily addressed by the latest round of revisions and heartily endorse publication. We are therefore delighted to inform you that your manuscript 'A data-driven characterisation of natural facial expressions when giving good and bad news' has been provisionally accepted for publication in PLOS Computational Biology!

Before your manuscript can be formally accepted, some formatting changes are required - you will receive instructions on how to apply these in a follow up email and a member of our team will be in touch with a set of requests. Please note that your manuscript will not be scheduled for publication until you have made the required changes, so a swift response is appreciated.

Congratulations again! 

Rachael Jack, PhD

Guest Editor

PLOS Computational Biology

Wolfgang Einhäuser

Deputy Editor

PLOS Computational Biology

Reviewer's Responses to Questions

**Comments to the Authors:**

Reviewer #1: The authors have addressed all of my concerns and I am now happy to recommend publication

Reviewer #2: This is a very interesting paper that applies a novel, data-driven method to investigate whether natural occurring facial expressions can be distinguished when these are accompanied by positive and negative communications. I was already enthusiastic about this paper when I read it the first time and the authors have done a very good job in answering my questions, providing further clarifications, and dealing with all of my comments. I very much enjoyed reading it and I have no further questions or comments.

Reviewer #3: I have no further comments.

**Have all data underlying the figures and results presented in the manuscript been provided?**

Reviewer #1: None

Reviewer #2: Yes

Reviewer #3: Yes

PLOS authors have the option to publish the peer review history of their article (what does this mean?). If published, this will include your full peer review and any attached files.

Reviewer #1: No

Reviewer #2: No

Reviewer #3: No

---

## [Editor Report · Acceptance letter]

19 Oct 2020

PCOMPBIOL-D-20-00591R1 

A data-driven characterisation of natural facial expressions when giving good and bad news

Dear Dr Watson,

I am pleased to inform you that your manuscript has been formally accepted for publication in PLOS Computational Biology. Your manuscript is now with our production department and you will be notified of the publication date in due course.

With kind regards,

Laura Mallard
